# High-performance and scalable on-chip digital Fourier transform spectroscopy

Derek M. Kita [1,2], Brando Miranda[3], David Favela[4], David Bono[1], Jérôme Michon[1,2], Hongtao Lin [1,2], Tian Gu[1,2] & Juejun Hu[1,2]

On-chip spectrometers have the potential to offer dramatic size, weight, and power advantages over conventional benchtop instruments for many applications such as spectroscopic sensing, optical network performance monitoring, hyperspectral imaging, and radio-frequency spectrum analysis. Existing on-chip spectrometer designs, however, are limited in spectral channel count and signal-to-noise ratio. Here we demonstrate a transformative on-chip digital Fourier transform spectrometer that acquires high-resolution spectra via time-domain modulation of a reconfigurable Mach-Zehnder interferometer. The device, fabricated and packaged using industry-standard silicon photonics technology, claims the multiplex advantage to dramatically boost the signal-to-noise ratio and unprecedented scalability capable of addressing exponentially increasing numbers of spectral channels. We further explore and implement machine learning regularization techniques to spectrum reconstruction. Using an 'elastic-$D_1$' regularized regression method that we develop, we achieved significant noise suppression for both broad (>600 GHz) and narrow (<25 GHz) spectral features, as well as spectral resolution enhancement beyond the classical Rayleigh criterion.

[1] Department of Materials Science & Engineering, Massachusetts Institute of Technology, Cambridge, MA, USA. [2] Materials Research Laboratory, Massachusetts Institute of Technology, Cambridge, MA, USA. [3] Center for Brains, Minds & Machines, Massachusetts Institute of Technology, Cambridge, MA, USA. [4] Department of Mechanical Engineering, Massachusetts Institute of Technology, Cambridge, MA, USA. Correspondence and requests for materials should be addressed to D.M.K. (email: dkita@mit.edu) or to J.H. (email: hujuejun@mit.edu)

Optical spectrometers are extensively applied to sensing, materials analysis, and optical network monitoring. Conventional spectrometers are bulky instruments often involving mechanical moving parts, which severely compromises their deployment versatility and increases cost. Photonic integration offers a solution to miniaturize spectrometers into a chip-scale platform, albeit often at the cost of performance and scalability. To date, the majority of on-chip spectrometers have relied on dispersive elements such as gratings[1–7], holograms[8,9], and microresonators[10–12]. When applied to high-resolution spectrum acquisition, these devices suffer serious signal-to-noise ratio (SNR) penalties as a result of spreading input light over many spectral channels. Furthermore, the device footprint and system-level complexity increases linearly with the number of spectral channels $N$, since the spectral resolution scales inversely with the optical path length (OPL) and each channel requires a dedicated photodetector. The SNR degradation and linear scaling behavior preclude high-performance on-chip spectrometers with channel counts rivaling their benchtop counterparts, which typically have hundreds to thousands of spectral channels. We note that these constraints also apply to spectrometers based on the wavelength multiplexing principle, where each receiver captures an ensemble of monochromatic light rather than one single wavelength[13–19] (for multiple-scattering-based spectrometers, the device dimension scales with spectral resolution quadratically).

Unlike dispersive spectrometers, Fourier transform infrared (FTIR) spectrometers overcome the trade-off between SNR and spectral resolution benefiting from the multiplex advantage, also known as Fellgett's advantage[20]. Traditional benchtop FTIR spectrometers use moving mirrors to generate a tunable OPL, a design not readily amenable to planar photonic integration. On-chip FTIR spectrometers instead rely on thermo-optic or electro-optic modulation to change the OPL in a waveguide[21–24]. The miniscule refractive index modifications produced by these effects, however, result in a large device footprint and constrain the practically attainable spectral resolution to tens of cm$^{-1}$ in wave number, far inferior compared to their benchtop counterparts. Furthermore, prior work has demonstrated Fourier-transform spectrometers using arrays of discrete Mach–Zehnder interferometers (MZIs)[15,18,25,26], although these approaches are not practical for large spectral channel counts (due to excessive chip footprints). The compressive sensing methods proposed and demonstrated for reconstructing optical spectra using the limited measurements from these MZI arrays are also only practical for sparse spectra such as laser lines (as shown in later sections). Prior theoretical work has suggested optical switches can be used to physically alter interferometer arm lengths[27], but these embodiments would all suffer in practice from poor scaling laws, as the number of spectral channels is equal to the number of switches and photodetectors.

We propose and experimentally demonstrate a novel digital Fourier transform (dFT) spectrometer architecture that resolves the performance and scalability challenges of these prior approaches. Our device consists of a MZI with optical switches on each arm that direct light to waveguides of unique path lengths[28]. This approach claims three key advantages over state-of-the-art techniques. First, both the resolution and spectral channel count scale exponentially with $j$, the number of optical switches. The unique exponential scaling laws inherent to our dFT architecture enable high-resolution spectra to be acquired with minimal chip space and control electronics. Second, direct modification of the waveguide path offers over 2 orders of magnitude larger OPL modulation per unit waveguide length compared to thermo-optic- or electro-optic-based index modulation (see Supplementary Note 3 for further details), enabling superior spectral resolution within a compact device. Third, the device benefits from the multiplex advantage (which ensures a higher SNR over dispersive type

spectrometers) and requires only a single-element photodetector rather than a linear detector array, which reduces cost and system complexity.

## Results

**dFT architecture and scaling laws.** The dFT spectrometer comprises of a reconfigurable MZI, which is schematically illustrated in Fig. 1a. Each arm of this interferometer consists of $j/2$ cascaded sets of optical switches (where $j$ is an even integer) connected by waveguides of varying lengths. An OPL difference of zero between the two arms is achieved when light is directed to the set of reference paths (marked with black color) in both MZI arms. Lengths of the waveguide paths in red differ from the reference paths by powers of two times $\Delta L$. In this device architecture, each permutation of the switches corresponds to a unique OPL difference between the arms that covers 0 to $(2^j - 1) \cdot n_g \cdot \Delta L$ with a step size of $n_g \cdot \Delta L$, where $n_g$ represents the waveguide group index. In traditional FTIR spectrometers the OPL difference between the two arms is continuously tuned, but in our dFT spectrometer the state of each "digitized" binary optical switch corresponds to a unique permutation of the spectrometer and a unique OPL difference. The number of spectral channels, defined by the distinctive optical states the device furnishes, is:

$$N = 2^j \tag{1}$$

and the spectral resolution is given following the Rayleigh criterion[29–31]:

$$\delta\lambda = \frac{\lambda^2}{(2^j - 1) \cdot n_g \Delta L} \approx \frac{1}{2^j} \cdot \frac{\lambda^2}{n_g \Delta L} \tag{2}$$

where $\lambda$ denotes the center wavelength.

**Experimental device.** We experimentally validated the dFT spectrometer concept by demonstrating a 64-channel device ($j = 6$) operating at the telecommunication C-band. The device was fabricated leveraging a commercial silicon photonics foundry process, where the optical switches employ a custom compact thermo-optic phase shifter design (89.5 µm long, with 33 mW/π phase shifting efficiency and 30.8 µs 10–90% rise time, both evaluated through our own experimental measurements)[32]. The insertion loss of the entire spectrometer was found to be 9.1 ± 1.7 dB, with a loss of 1.7 ± 0.4 dB for each switching stage (averaged across the full 20 nm band considered in this text, see Supplementary Note 2). Figure 1b presents a micrograph of the spectrometer after front-end-of-line silicon fabrication. The chip was subsequently packaged with bonded fiber arrays and electrical connections. Details of the fabrication and packaging processes are elaborated in the Methods section. The spectrometer also integrates an on-chip germanium photodetector and a standard FC/PC fiber connector interface, making it a standalone "plug-and-play" device for optical spectrum analysis (Fig. 1c). The spectrometer dissipates 99 mW of power during operation, regardless of the switch state.

The spectrometer was characterized using a setup depicted in Fig. 2a. High-resolution transmittance spectra of the device were first recorded by wavelength sweeping a tunable laser between 1550 and 1570 nm, for all 64 permutations of the switch on/off combinations. The 64 spectra (which show an average extinction ratio of 21 ± 3 dB) are plotted in Fig. 2c, each associated with a unique OPL difference between the MZI arms. The ensemble of spectra forms an $m \times n$ calibration matrix **A**. Each row of **A** represents a transmittance spectrum and contains $n = 801$

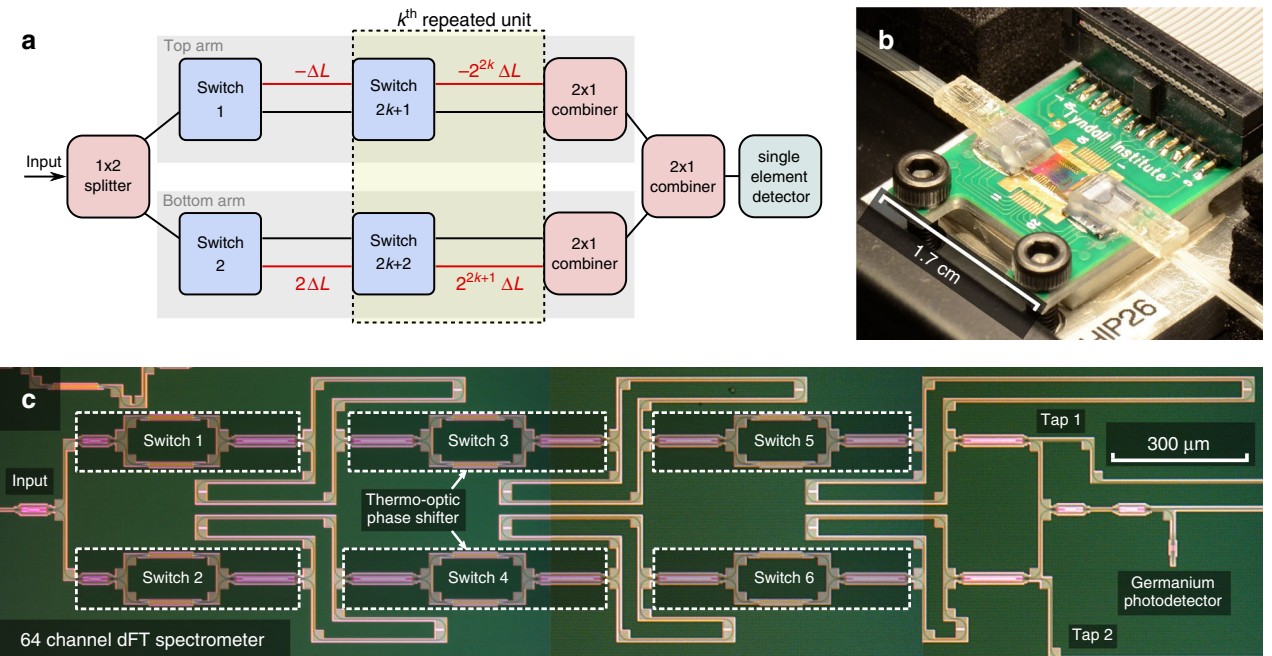

**Fig. 1** Images and schematics of the dFT architecture. **a** Block diagram illustrating the generic structure of a dFT spectrometer with *j* switches and $K=j/2-1$ repeated stages indexed by $k \in [1, K]$; **b** photo of the fully packaged, plug-and-play dFT spectrometer with standard FC/PC fiber interface and a ribbon cable for control and signal read-out; **c** top-view optical micrograph of the 64-channel dFT spectrometer after front-end-of-line silicon fabrication, showing the interferometer layout, the thermo-optic switches and waveguide-integrated germanium photodetector

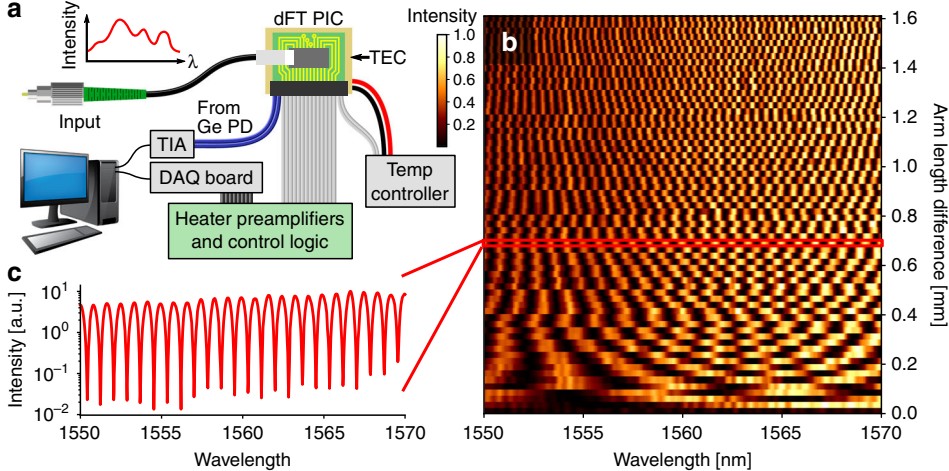

**Fig. 2** Packaged dFT photonic integrated circuit (PIC) and spectral basis set. **a** Schematic diagram of the dFT spectrometer characterization setup; **b** an exemplary transmission spectrum of the dFT device corresponding to an arm length difference of 0.7 mm; **c** transmission spectra of the device for all 64 permutations of the switch on/off combinations: the ensemble of 64 spectra constitute the basis set for spectrum reconstruction

elements, the number of wavelength points in the scan. Each column corresponds to a discretely sampled interferogram of the narrow-band laser and contains $m = 64$ elements. The intensity measured by the detector for an arbitrary input polychromatic signal (represented by a column vector **x** with 801 elements) is:

$$\mathbf{y} = \mathbf{Ax} \qquad (3)$$

where the interferogram **y** is a column vector with 64 elements, each gives the detector output at a particular switch permutation. The vector **y** was measured by recording the detector output at all 64 permutation states. Since we measured **y** with size 64

to infer **x** with size 801, the system is underconstrained and therefore regularization techniques are required to specify a unique solution.

**Spectrum reconstruction**. To determine the correct solution vector, we first explored a number of standard regularized regression and compressive-sensing techniques such as basis pursuit denoising (BPDN), least absolute shrinkage and selection operator (LASSO), and elastic-net—the corresponding metrics they optimize are listed in Table 1. For each of these regularized regression methods, a weight (hyperparameter) is applied to the

$L_1$-norm (and for elastic-net a separate weight for the $L_2$-norm) of the spectrum. To determine the appropriate value(s) of the hyperparameter(s), we use a standard holdout cross-validation technique[33], through which we take two consecutive measurements of the interferogram ($\mathbf{y_1}$ and $\mathbf{y_2}$) and choose hyperparameters that maximize the coefficient of determination $R^2(\mathbf{A_2x}, \mathbf{y_2})$ (see Supplementary Note 1), where $\mathbf{A_1}$ and $\mathbf{A_2}$ are two separate measurements of the basis (performed once as a calibration step), and $\mathbf{x}$ is the computed spectrum from $\mathbf{y_1}$ and $\mathbf{A_1}$. We note that our basis measurements $\mathbf{A_1}$ and $\mathbf{A_2}$ do not change significantly over time: in fact, for all results reported in this work the calibration matrices $\mathbf{A_1}$ and $\mathbf{A_2}$ are measured more than 2 weeks prior to the actual interferogram measurements $\mathbf{y_1}$ and $\mathbf{y_2}$. The validated device stability is critical to practical applications, as it ensures that the calibration step only need to be performed once for each spectrometer module. We found that these methods (BPDN, LASSO, elastic-net) tend to perform reasonably well for sparse spectra such as a few input laser lines. However, BPDN and LASSO (which only weight the $L_1$-norm) fail to faithfully reproduce broad spectral features where the input spectrum contains few (if any) zeros, as illustrated by the low $R^2$ values in Table 2 as well as examples of the reconstructed spectra presented in Supplementary Fig. 1 and Supplementary Fig. 2. Other techniques we considered, such as Ridge Regression (which only weights the $L_2$-norm), pseudo-inverse (Penrose–Moore), and radial-basis function (RBF) networks, exhibit significant

reconstruction errors on both broad and narrow spectral features. The latter method (RBF Network) is the analytical solution to a regression model that bounds all derivatives of the solution vector[34].

To properly account for both the sparsity and "smoothness" of the spectra, we implemented a regularized regression model that accounts for both $L_1$- and $L_2$-norms as well as the first-derivative of the spectrum. This method, that we call "elastic-$D_1$" from here on, is a non-negative elastic-net method with an additional smoothing prior. It solves the regularization problem:

$$\min_{\mathbf{x},\mathbf{x}>0}\left\{||\mathbf{y}-\mathbf{Ax}||_2^2 + \alpha_1||\mathbf{x}||_1 + \alpha_2||\mathbf{x}||_2^2 + \alpha_3||\mathbf{D_1x}||_2^2\right\} \quad (4)$$

where $\alpha_1$, $\alpha_2$, and $\alpha_3$ are hyperparameters that weight the corresponding $L_1$- and $L_2$-norms on $\mathbf{x}$, and the $L_2$-norm on the first derivative specified by the matrix $\mathbf{D_1}$. The combination of bounds on the $L_1$-norm (that induces sparsity on the spectrum), $L_2$-norm (that bounds magnitude of the spectrum), and first derivative of the spectrum (that sets the desired smoothing) produce significantly better reconstructions (compared to other known methods) on both broad and narrow spectral features without requiring knowledge of the true input spectrum. Since Eq. (4) is a non-negative quadratic program, it is readily solvable with standard convex optimization tools[35]. Details of the elastic-$D_1$ algorithm, its implementation, and performance benchmarking compared to several existing reconstruction techniques are presented in Table 2 and Supplementary Note 1. The comparison in Table 2 suggests that the elastic-$D_1$ algorithm significantly outperforms other techniques for both sparse and broadband input signals.

To demonstrate the versatility of the elastic-$D_1$ method, we applied the technique to experimentally measured interferograms for two types of polychromatic inputs, sparse signals consisting of discrete laser lines and broadband signals with complex spectral features (see Methods). Figure 3 plots the reconstructed spectra comprising two laser lines with slightly different amplitudes and varying wavelength spacing. The elastic-$D_1$ technique precisely reproduces the laser wavelengths with ±0.025 nm accuracy, only limited by the finite wavelength step size of the calibration matrix (0.025 nm). The spectral resolution of our device, determined here by the minimum resolvable wavelength detuning between two laser lines, significantly outperforms the Rayleigh criterion of 0.4 nm with an experimentally determined value of 0.2 nm. The enhanced reconstruction quality is a result of the elastic-$D_1$ method's automatic consideration of the tradeoffs between spectral sparsity, magnitude, and smoothness (see Supplementary

## Table 1 Reconstruction methods considered

| Method | Problem |
|---|---|
| Pseudoinverse | $\mathbf{y}=\mathbf{Ax}$ (Moore–Penrose) |
| Ridge Regression | $\min_x\left\{||\mathbf{y}-\mathbf{Ax}||_2^2 + \alpha_2||\mathbf{x}||_2^2\right\}$ |
| LASSO | $\min_x\left\{||\mathbf{y}-\mathbf{Ax}||_2^2 + \alpha_1||\mathbf{x}||_1\right\}$ |
| BPDN | $\min_x\left\{(1/2)||\mathbf{y}-\mathbf{Ax}||_2^2 + \alpha_1||\mathbf{x}||_1\right\}$ |
| RBF Network | $\min_c\left\{||\mathbf{y}-\mathbf{Ah_c}||_2^2\right\}$ with $\mathbf{h_c}=\mathbf{Kc}=\sum_{d=1}^{D}c_d e^{-\beta|\lambda-\lambda_d|^2}$ |
| Elastic-Net | $\min_{x,x>0}\left\{||\mathbf{y}-\mathbf{Ax}||_2^2 + \alpha_1||\mathbf{x}||_1 + \alpha_2||\mathbf{x}||_2^2\right\}$ |
| Elastic-$D_1$ | $\min_{x,x>0}\left\{||\mathbf{y}-\mathbf{Ax}||_2^2 + \alpha_1||\mathbf{x}||_1 + \alpha_2||\mathbf{x}||_2^2 + \alpha_3||\mathbf{D_1x}||_2^2\right\}$ |

Spectral reconstruction techniques/methods considered in this work, and the corresponding problem they solve. Depending on the nature of the problem and input vector, various techniques are such as convex optimization and gradient descent are available to solve the problem. The c coefficients for the RBF Network are computed via $\mathbf{c}=(\mathbf{AK})^+\mathbf{y}$, where $\mathbf{K}$ is the kernel matrix $K_{\lambda,\lambda_d}=e^{-\beta|\lambda-\lambda_d|^2}$ and $\lambda_d$ are the centers of the radial basis functions

## Table 2 Performance comparison of reconstruction methods

| Method | Mean $R^2$ value (sparse signals) | $R^2$ value (Fig 4A) | $R^2$ value (Fig 4B) | $R^2$ value (Fig 4C) | Mean FPCT [msec] | Hyper-parameter search space |
|---|---|---|---|---|---|---|
| Pseudoinverse | $-10.709 \pm 2.438$ | 0.763 | 0.324 | 0.636 | < 0.01 | 1 |
| Ridge Regression | $-10.901 \pm 2.344$ | 0.710 | 0.327 | 0.717 | 1.08 | 500 |
| LASSO | $0.975 \pm 0.035$ | 0.098 | $-0.225$ | $-0.247$ | 29.08 | 500 |
| BPDN | $0.946 \pm 0.09$ | $-0.351$ | $-0.687$ | $-0.388$ | 155.0 | 500 |
| RBF Network | $-10.705 \pm 2.455$ | 0.766 | 0.337 | 0.650 | 47.63 | 200 |
| Elastic-Net | $0.992 \pm 0.012$ | 0.848 | $-0.220$ | 0.632 | 22.6 | $200 \times 200$ |
| **Elastic-$D_1$** | **$0.954 \pm 0.057$** | **0.912** | **0.820** | **0.834** | **291.0** | **$12 \times 12 \times 12$** |

Average and standard deviation of $R^2$-values for measurements of sparse spectra consisting of a single laser line at 11 different wavelengths evenly spaced on the range $\lambda$=[1555 nm, 1565 nm] and broadband spectra (black curves in Fig. 4). For each reconstruction technique, we include the average fixed-parameter compute time (FPCT) required to solve the corresponding problem (shown in Table 1) and the dimensionality/size of the hyperparameter space used. Calculations were all performed on a laptop with an Intel Xeon E3-1505M v5 CPU and 16 GB of RAM. Blue shaded boxes correspond to $R^2$ values above 0.8, yellow is for $R^2$ between 0.6 and 0.8, and red is for $R^2$ less than 0.6. The results indicate that the elastic-$D_1$ algorithm consistently and accurately reconstructs both sparse and broadband input signals

Note 1). Figure 4 compares the spectra of three unique broadband inputs recorded using a benchtop optical spectrum analyzer as a reference and reconstructed spectra using the same 64-channel dFT device and elastic-$D_1$ algorithm. The high reconstruction quality on arbitrary input spectra with characteristic spectral features ranging from several nanometers to well below the Rayleigh limit validates dFT spectroscopy as a generic, powerful tool for quantitative spectroscopy.

## Discussion

Optical spectrum analysis using a MZI with physical path differences enables high-resolution spectroscopy with limited interferogram measurements. By utilizing a reconfigurable interferometer, we demonstrated a working dFT spectrometer device that both recovers the multiplex advantage (as seen by the low-insertion loss per channel in Supplementary Table 1 and Supplementary Note 5), and is capable of high-resolution spectrum acquisition with a compact form factor. For input spectra with slowly varying (broad) spectral features, our elastic-$D_1$ spectral reconstruction provides a superior reconstruction accuracy than other regularization or compressive sensing techniques at the expense of increased computation time to determine suitable hyperparameters. A thorough comparison of the elastic-$D_1$ technique compared to other common reconstruction methods is shown in Supplementary Table 1 and graphically depicted in Supplementary Fig. 1 and Supplementary Fig. 2. We note that noticeable errors tend to occur at the edges of broad spectra after reconstruction since the signal (interferogram) is bandlimited in the spatial domain. However, these errors are expected to become negligible as the number of unique OPL differences (determined by the number of switching stages via Eq. (1)) and spectral channels increases. As channel count increases, the system also becomes less underconstrained and reconstruction errors are expected to decrease significantly. Increasing channel count is simple and inexpensive in terms of chip-space—a dFT spectrometer with two additional (5 total) switching stages (i.e., a total of $j = 10$ switches) would access 1024 unique spectral channels while occupying only 40% more chip area than the 64-channel device demonstrated here. We believe this device architecture paves the way toward future on-chip spectrometers with significantly larger spectral channel counts, higher resolution, and dramatically increased optical throughput with respect to existing on-chip spectrometers.

In conclusion, this work pioneers dFT spectroscopy as a high-performance, scalable solution for on-chip optical spectrum analysis. Its unique exponential scalability in performance, superior SNR leveraging the multiplex advantage, as well as compact and remarkably simplified device design are among the key advantages of the technology. Moreover, its proven compatibility with industry-standard foundry processes enables scalable manufacturing and drastic cost reduction. We further

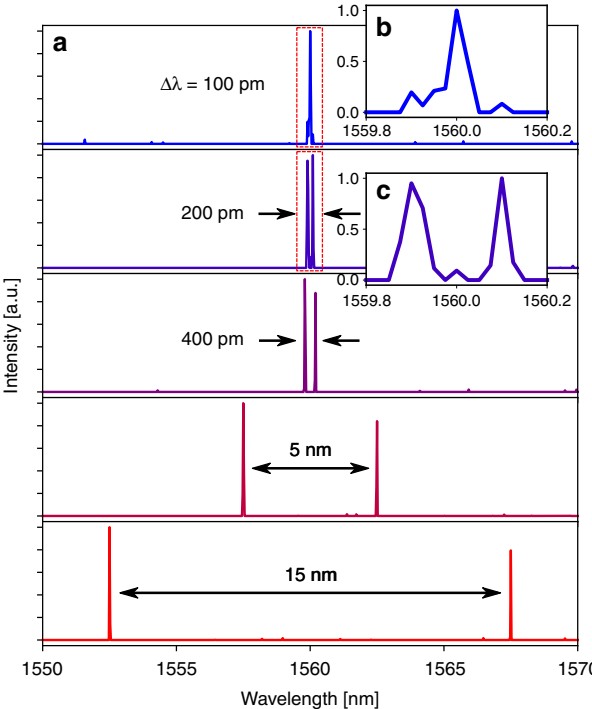

**Fig. 3** Sparse signal reconstruction. **a** Spectra consisting of two laser lines with varying spacing measured using the 64-channel dFT spectrometer and reconstructed by applying the elastic-$D_1$ algorithm. Insets **b** and **c** show zoomed-in images of the narrow spectral features for input laser lines with 100 and 200 pm spacing, respectively

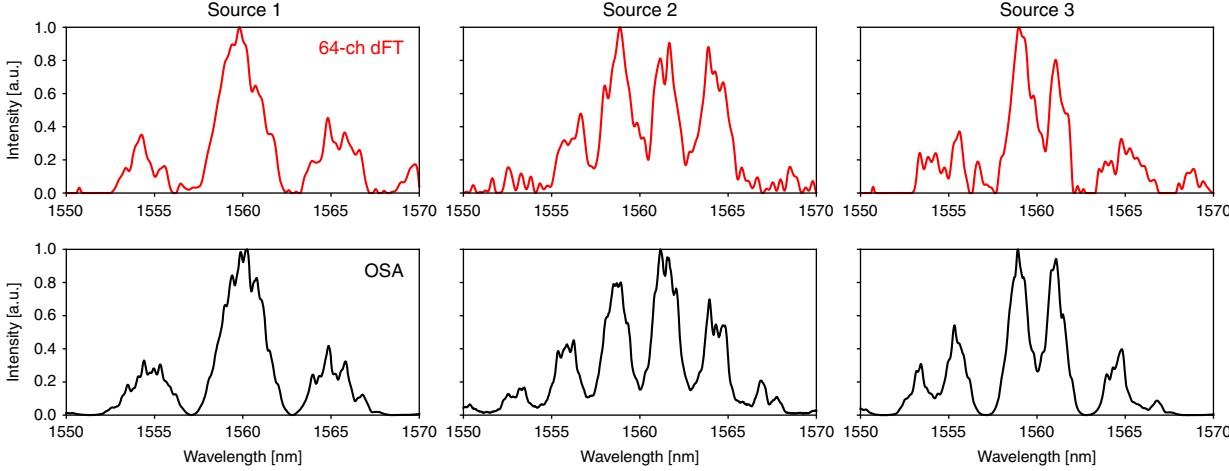

**Fig. 4** Broadband signal reconstruction. Three unique light sources with broad spectral features were measured using the dFT spectrometer chip and elastic-$D_1$ method (top red curves) and compared to measurements by a benchtop optical spectrum analyzer (OSA) (bottom black curves). The three light sources were generated using amplified spontaneous emission from an erbium-doped fiber passed through several Mach–Zehnder interferometers (additional information is provided in the Methods)

developed an elastic-$D_1$ machine learning regularization technique to achieve significant noise suppression and resolution enhancement. The powerful combination of dFT spectroscopy and machine learning techniques will empower future applications of spectroscopy such as chemical and biological sensors-on-a-chip, space-borne spectroscopy, optical network monitoring, and radio-frequency spectrum analysis.

## Methods

**Device fabrication.** Device layout and mask generation was done using Luceda's IPKISS design framework and tools. The dFT spectrometer chips were fabricated on the imec-ePIXfab active silicon photonics platform (ISIPP25G) multi-project wafer service, using 193 nm deep-UV lithography. Standard passive and active components (excluding the custom compact thermo-optic phase modulator) from the imec-ePIXfab process design kit (PDK) library were used to construct the dFT spectrometer (schematics shown in Supplementary Fig. 3). The devices were subsequently packaged at Tyndall National Institute with fiber grating coupler arrays and electrical connections on a thermoelectric cooler for temperature control.

**Device characterization.** Packaged devices were characterized at MIT with a swept single-frequency external cavity laser to determine the wavelength response of the device for each switch state, using the integrated germanium detector for signal readout and temperature stabilization at $25.00 \pm 0.01$ °C (for the device's temperature dependence, see Supplementary Note 4 and Supplementary Fig. 4). The thermo-optic switches were initially calibrated by tuning the heater powers until the frequency response of the top arm or bottom arm was flat, as measured from one of two "tap" ports, indicated in Fig. 1. Heater preamplifiers, programmable digital control logic, and photodetector transimpedance amplifiers with variable gain are all implemented with custom electronics and automation software. Each interferogram measurement took 2.7 seconds to acquire (limited primarily by the USB latency between our computer and DAQ board). The elastic-$D_1$ technique was implemented in Python 2.7 with free software for convex optimization (cvxopt[35]). The Ridge Regression, LASSO, RBF network, and elastic-net methods were implemented using scikit-learn, a free library for machine learning in Python[36]. The basis-pursuit denoising method was implemented using SPORCO, a free Python library[37]. Testing on two input-lasers with different wavelength spacing was performed by combining two separate tunable continuous-wave external cavity lasers through a $2 \times 2$ fiber coupler with one output port to an optical spectrum analyzer and the second to the dFT spectrometer. To generate the broadband input signal shown in Fig. 4 source 1 and source 2, we couple the amplified spontaneous emission from an erbium-doped fiber amplifier (EDFA) to on-chip imbalanced MZI structures with arm length differences of 100 and 200 μm, respectively. The spectrum in Fig. 4 source 3 is obtained by passing light through both MZI structures connected in series. The complex spectral features are attributed to both interference in the imbalanced MZI as well as Fabry–Perot fringes due to multiple reflections at connectors and couplers. The reference spectra were recorded using a Yokogawa AQ6375B optical spectrum analyzer.

**Code availability.** Custom code written in Python to perform spectral reconstruction via the elastic-$D_1$ regularized regression technique is available upon reasonable request.

## Data availability

Raw data from the basis measurements and all interferogram measurements are available from the corresponding authors upon reasonable request.

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

## Acknowledgements

The authors thank Lionel C. Kimerling, Anu Agarwal, and Rajeev Ram for providing access to device measurement facilities. Funding support is provided by the National Science Foundation under award number 1709212, MIT SENSE.nano Seed Grant, and the Department of Energy under Grant DE-NA0002509. D.M.K. acknowledges the Saks-

Kavanagh Fellowship for Technology Commercialization provided by the Saks-Kavanaugh Foundation for financial support.

## Author contributions

D.M.K. designed and characterized the spectrometer device. B.M. and D.M.K. developed the machine learning algorithms for spectrum reconstruction. D.F., D.B., and D.M.K. designed and assembled the electronics for device testing. J.M. and H.L. assisted in device characterization. J.H. conceived the spectrometer concept and supervised the research. T. G. and D.M.K. contributed to the concept formulation. All authors contributed to technical discussions and writing the paper.

## Additional information

**Competing interests:** The authors declare no competing interests.

