## [Peer Review File · Nature Communications]

Reviewers' comments:

Reviewer #1 (Remarks to the Author):

The paper reports a novel spectrometer architecture that overpasses existing on-chip spectrometer devices, in term of the fundamental trade-off between resolution and compactness/complexity. Furthermore a novel machine learning technique is also developed for spectrum reconstruction, that produces better reconstruction on both broadband and narrow spectral features. The results correspond to significant advances in term of on-chip spectroscopy for different applications in sensing or spectroscopy, and can be translated to different material and wavelength domains. Comments and suggestions are reported below:

- When comparing the OPL achieved from the dFT with more classical structures in which the modification of the waveguide path is obtained by thermo optic or electro-optic effect, the authors mention an increase of the OPL variation by a factor of 100 for the dFT. This value can seem arbitrary, as large OPL variation can also be obtained if the phase of long waveguides (few cms long) are tuned. As this would be at the expense of the power consumption, the authors could mention the difference of power consumption for a given OPL variation between the 2 approaches (for example mentioning the power that would be required from a thermal heater to obtain an OPL variation of a 0.7mm, compared with the power consumption of the optical switch used in the reported work).
- It is written that "the device is far less sensitive to temperature variations than existing on-chip FTIR spectrometers, as the temperature-induced OPL fluctuation scales linearly with the physical length of the interferometer arms". This argument should be more justified. Indeed from my understanding, the OPL differences used with the dFT spectrometer are the same as the ones obtained with a Fourier Transform spectrometer based on an array of discrete Mach Zehnder interferometer. The main advantage of the dFT is that these OPL variations are obtained by turning ON and OFF the optical switches. In this case, the difference in term of sensitivity to temperature fluctuations could be made clearer.
- Furthermore in the experimental set-up in Figure 2, it is indicated that a temperature controller is required to control the chip temperature. Could the authors comment the maximum tolerable temperature variation of the chip to maintain good operation ?
- In the discussion the authors evaluate the performance of a 5 switching stages dFT spectrometers. Could they also comment on the value of the length difference which is 700 μm in the experimental demonstration ? Could the dFT be combined with spiral waveguides to increase the resolution in a compact devices ?

Reviewer #2 (Remarks to the Author):

Authors described a Fourier-Transform spectrometer architecture and an algorithm to retrieve spectral information in a better way. I have some comments and suggestions to the authors based on the claims they made and the results they showed.

1. Firstly, the spectrometer architecture given in this manuscript is not completely novel and it has been published by the authors in a different journal a year ago. There is also a patent application related with this design. However, most importantly, there is a publication from 2005 with a similar multi-stage delay line approach for a different application [1]. Additionally, an alternative FT-spectrometer design proposal published in 2017 which addresses the penalties of existing FT-spectrometers; the main motivation behind this manuscript [2]. Later, an improved layout for broadband applications using a single detector approach was discussed in a book chapter [3]. Authors overlooked these articles/book chapters and did not cite them in their manuscript. In this

sense, the novelty of this work mainly comes from its spectrum reconstruction algorithm in contrary to what they claimed in the Introduction part.

[1] "Integrated-optic variable delay line and its application to a low-coherence reflectometer," Opt Lett. 2005 Oct 15;30(20):2739-41.

[2] "Design of a compact and ultrahigh-resolution Fourier-transform spectrometer," Optics Express, Vol. 25, Issue 2, pp. 1487-1494, (2017).

[3] Advances in Optics: Reviews. Book Series, Vol. 3, Chapter 9

2. What are the dimensions of the optical waveguides used in the spectrometer layout? Rib/ridge waveguide or channel waveguide used? Why did you specifically use silicon material platform? Compactness is an important aspect, however considering the penalties of Silicon photonics in terms of loss and back reflections due to sidewall roughness, it does not give the best performance. The motivation of waveguide/material platform as well as some crucial information that could make this research repeatable for other researchers are missing. Is this due to the fact that there is a commercial interest in this device?

3. What are the sources of 2.75 dB loss per stage? Please also talk about the possible improvements if there is any.

4. How do your thermo-optic switch, 1x2 splitter, 2x1 combiner look like? Please include drawings together with some simulation results to show how much loss is expected in theory from these components. I am assuming that you used Y-splitters mainly, however did you reduce their losses by optimizing the geometry if not a e-beam writing method was used in the fabrication?

5. Even though the layout given here looks simple, I believe that finding the right OPL values by perturbing the phase values of each thermo-optic switch is laborious. What is the typical time needed to reconstruct spectral information, such as, given in Fig. 4?

Reviewer #3 (Remarks to the Author):

This paper demonstrates for the first time the implementation of an on-chip Fourier-transform spectrometer using integrated thermo-optic switches, which are used to reconfigure the circuit to have different optical path lengths at different times. Because of the size, weight, and power advantages of on-chip spectrometers compared to bench top systems they have the potential to find application in many areas.

The critical advantage of this new spectrometer compared to existing on-chip spectrometer designs is that this architecture is easily scalable to very high spectral channel counts with limited corresponding increase of the on-chip footprint of the circuit, which is extremely important if on-chip spectrometers are to be used widely. There are several additional advances presented here, including presentation of a novel spectral retrieval procedure, integration of the spectrometer with on-chip detectors, and packaging with interface electronics.

I believe that this work can have a high impact in the field, that it is technically strong, and that evidence is provided for most of its conclusions. A good level of detail is provided, and it should be possible for another researcher to reproduce this work.

I have several questions and comments that I would like to see addressed before publication:

1. On page 2, paragraph 2 it is stated that:

"Furthermore, prior work has demonstrated Fourier-transform spectrometers using arrays of discrete Mach-Zehnder interferometers (MZIs)^{15,18,25,26}, although these approaches are not practical for large spectral channel counts (due to excessive chip footprints) and suffer from the same SNR penalties as dispersive-type spectrometers since power is evenly split into each of the N spectral channels."

The bold part of the sentence is potentially misleading, as it has previously been argued that for

these types of devices (e.g. Florjanczyk et al. "Multiaperture planar waveguide spectrometer formed by arrayed Mach-Zehnder interferometers" Optics Express, vol. 15, issue 26, 2007) a single input waveguide is used only to simplify device characterisation, and that a separate input waveguide could be used for each MZI. This would avoid the SNR penalties from splitting into N spectral channels, and would in fact increase the etendue, allowing more light to be coupled into the chip from spatially extended sources. Please could the bold text be amended?

2. What is the optical insertion loss of the spectrometer, and how much does it vary across the different interferometer channels?

3. Could the authors please comment on the visibility (i.e. extinction ratio) of the interferometer fringes, and on what the consequences are for the sensitivity or dynamic range of the spectrometer?

4. How long does the measurement of a single spectrum take (excluding additional averaging time)?

5. Since for some applications a wide spectrometer bandwidth would be crucially important, could comments be added to the paper to explain what limits the bandwidth of this current device, as well as what are the fundamental bandwidth limitations of a device of this kind?

6. On p. 4, paragraph 1, it is stated that "An additional benefit is that the device is far less sensitive to temperature variations than existing on-chip FTIR spectrometers, as the temperature-induced OPL fluctuation scales linearly with the physical length of the interferometer arms".

Could the authors please be more specific about which types of existing spectrometers they are referring to? It is my understanding that for a specific resolution and bandwidth the device presented here would require the same OPL as MZI array type devices (e.g. ref 2, 15, 18), and that therefore the sensitivity to thermal variations would be the same. Would any strategies be required in future to compensate for thermal variations?

7. Please explain more clearly the difference between the "Average fixed parameter compute time" and "Hyper-parameter search time" in the caption to Table 2.

8. Since compute times are given for the different reconstruction methods, please provide some details of the computer that was used for these calculations.

9. How much variation is there in the R^2 values given in table 2 when the reconstruction methods are tested on different broadband spectra (e.g. from figures 4A and 4B) for the broad spectra, or when the lasers are tuned to different wavelengths for the sparse spectra? In other words, are the R^2 values presented in table 2 the "best case" for the Elastic-D1 method? Would average R^2 values for different input spectra be more appropriate?

10. In the supplementary materials, on page 1, paragraph 5, it is stated that "...we use a standard holdout cross-validation technique, which requires only two successive measurements of the interferogram, characterized by the same input signal with different noise."

Would taking more than two successive measurements improve the accuracy of the calibration?

Response to Reviewer Comments NCOMMS-18-13737

Reviewer #1 (Remarks to the Author):

The paper reports a novel spectrometer architecture that overpasses existing on-chip spectrometer devices, in term of the fundamental trade-off between resolution and compactness/complexity. Furthermore a novel machine learning technique is also developed for spectrum reconstruction, that produces better reconstruction on both broadband and narrow spectral features. The results correspond to significant advances in term of on-chip spectroscopy for different applications in sensing or spectroscopy, and can be translated to different material and wavelength domains. Comments and suggestions are reported below:

- When comparing the OPL achieved from the dFT with more classical structures in which the modification of the waveguide path is obtained by thermo optic or electro-optic effect, the authors mention an increase of the OPL variation by a factor of 100 for the dFT. This value can seem arbitrary, as large OPL variation can also be obtained if the phase of long waveguides (few cms long) are tuned. As this would be at the expense of the power consumption, the authors could mention the difference of power consumption for a given OPL variation between the 2 approaches (for example mentioning the power that would be required from a thermal heater to obtain an OPL variation of a 0.7mm, compared with the power consumption of the optical switch used in the reported work).

Response:

We thank the reviewer for his/her comments. Indeed, we see how the factor of 100 can seem arbitrary to the reader. In this instance, we had been referring to the modulation efficiency (radians per unit length along the waveguide) for the three phase-shifting mechanisms: thermo-optic phase shifting, electro-optic phase shifting, and the amount of phase light naturally accumulates as it propagates in a waveguide. In the thermo-optic case, we assumed a constant temperature rise of 60°C in the Si waveguide (with $\frac{dn}{dT} = 1.8 \times 10^{-4} \text{ K}^{-1}$), and in the electro-optic case we assumed a typical $V_{\pi}L_{\pi} = 1 \text{ V}\cdot\text{cm}$ and applied voltage of 100 V. This gives a modulation efficiency of 433 rad/cm for the thermo-optic modulator and 314 rad/cm for the electro-optic modulators. In the direct path-modulation case we have a group index of $n_g = 4.25$ for the SOI waveguide geometry we employed, which yields a modulation efficiency of 172,281, a factor of 398 and 548 times greater than that of the thermo-optic and electro-optic shifters, respectively (100 was stated in the manuscript as a modest estimate).

We believe the proposed metric by the reviewer (power required for a given OPL variation) is in fact a good measurement for thermo-optically or electro-optically tuning an interferometer OPL. However, when directly modifying the path of light (switching it from a short path to a long path) in an interferometer, the switching power is fixed but the desired OPL is determined by the lithographic mask. Our device dissipates 99 mW of power regardless of what state it is in (path-switching turns one heater off and another on, as shown in the block-diagram schematic Figure S2 in the updated Supplemental Information). For this reason, it's difficult to compare our spectrometer to a heated interferometer, as we could arbitrarily increase our OPL for a given power consumption by lithographically defining longer waveguide arms. We have updated the manuscript to mention this 99 mW of power during operation. We have also provided a more detailed explanation of the metric mentioned above (phase shift per unit waveguide length) in the

manuscript that we hope will be helpful for comparing our demonstrated dFT technology with prior work on thermo-optic or electro-optic Mach-Zehnder interferometers.

- It is written that “the device is far less sensitive to temperature variations than existing on-chip FTIR spectrometers, as the temperature-induced OPL fluctuation scales linearly with the physical length of the interferometer arms”. This argument should be more justified. Indeed from my understanding, the OPL differences used with the dFT spectrometer are the same as the ones obtained with a Fourier Transform spectrometer based on an array of discrete Mach Zehnder interferometer. The main advantage of the dFT is that these OPL variations are obtained by turning ON and OFF the optical switches. In this case, the difference in term of sensitivity to temperature fluctuations could be made clearer.

Response:

Thank you for this response. This particular sentence is referring to the difference in sensitivity to temperature nonuniformity or thermal gradient (rather than a uniform temperature change) across a chip of our dFT spectrometer (which uses physical path length differences) to that of a thermo-optic or electro-optic FT spectrometer rather than an array of MZIs. This decreased temperature sensitivity originates from: 1) the smaller footprint of the device which minimizes temperature difference across the structure; and 2) the reduced waveguide lengths required to achieve an equivalent spectral resolution, which decreases the thermally induced arm imbalance in the presence of a thermal gradient.

The reviewer is correct in that the temperature sensitivity of our dFT is equivalent to the temperature sensitivity of other FT spectrometers that are arrays of discrete MZI's. We have removed this sentence in the manuscript and we hope this resolves any potential confusion.

- Furthermore in the experimental set-up in Figure 2, it is indicated that a temperature controller is required to control the chip temperature. Could the authors comment the maximum tolerable temperature variation of the chip to maintain good operation?

Response:

We thank the reviewer for this important question. In order to achieve accurate spectral reconstructions, it is important that the basis vectors that map the spectrum to the measured interferogram are constant. Since the device is quite compact, we expect the temperature of both interferometer arms to change by similar amounts (as opposed to a thermal-gradient, which would create an unbalanced arm temperature). This temperature change modifies the silicon refractive index and also the OPL by an amount proportional to the thermo-optic coefficient and the physical path length difference.

We performed a set of experiments to quantify the effects of a temperature detuning on the spectrum reconstruction. Using a single frequency laser tuned at 1560.0 nm, we measured the interferogram at number of different temperatures from 25.00 °C to 23.87 °C and reconstructed the spectrum using the elastic- D_1 method. The qualitative “sharpness” of the peak and amount of noise was unaffected by the temperature change, but the position of the center wavelength was detuned to longer wavelengths. Using this data, we characterized the first-order temperature

sensitivity to be $-85.2 \pm 1.3 \text{ pm}/^\circ\text{C}$, as shown in the figure below. For broadband input signals, we expect the R^2 -value to be much more robust to temperature variations (since the overlap of the two spectra is less sensitive to small wavelength shifts). For future readers, this information has been added to the Supplementary Information section.

- In the discussion the authors evaluate the performance of a 5 switching stages dFT spectrometers. Could they also comment on the value of the length difference which is $700 \mu\text{m}$ in the experimental demonstration? Could the dFT be combined with spiral waveguides to increase the resolution in a compact devices?

Response:

The length difference δL between different dFT states was lithographically defined to be $25.554 \mu m$, which yielded a maximum path length difference between the arms of $\Delta L = 63 \times 25.554 \mu m = 1609.902 \mu m$. These particular values were selected such that a bandwidth of approximately $\frac{1}{n_g \delta L} \sim 2.75$ THz (or 20 nm bandwidth in the telecomm C-band) and resolution of roughly 400 pm could be theoretically obtained, according to the Rayleigh criterion (we give approximate values because the group index value was not known with good certainty prior to chip fabrication).

The reviewer is correct in noting that spiral waveguides may also be used in future designs to enable longer MZI arms (for higher spectral resolution) in a smaller chip space. This will be a design to be explored for our next iteration.

Reviewer #2 (Remarks to the Author):

Authors described a Fourier-Transform spectrometer architecture and an algorithm to retrieve spectral information in a better way. I have some comments and suggestions to the authors based on the claims they made and the results they showed.

1. Firstly, the spectrometer architecture given in this manuscript is not completely novel and it has been published by the authors in a different journal a year ago. There is also a patent application related with this design. However, most importantly, there is a publication from 2005 with a similar multi-stage delay line approach for a different application [1]. Additionally, an alternative FT-spectrometer design proposal published in 2017 which addresses the penalties of existing FT-spectrometers; the main motivation behind this manuscript [2]. Later, an improved layout for broadband applications using a single detector approach was discussed in a book chapter [3]. Authors overlooked these articles/book chapters and did not cite them in their manuscript. In this sense, the novelty of this work mainly comes from its spectrum reconstruction algorithm in contrary to what they claimed in the Introduction part.

[1] "Integrated-optic variable delay line and its application to a low-coherence reflectometer," Opt Lett. 2005 Oct 15;30(20):2739-41.

[2] "Design of a compact and ultrahigh-resolution Fourier-transform spectrometer," Optics Express, Vol. 25, Issue 2, pp. 1487-1494, (2017).

[3] Advances in Optics: Reviews. Book Series, Vol. 3, Chapter 9

Response:

We thank the reviewer for his/her comments. Although the generic dFT architecture that we show in this article was first theoretically described by our own prior work [4], this work differs in several key aspects that we believe make our work novel and of interest to a much broader audience. In particular, this work shows an experimental demonstration of the digital Fourier-transform spectrometer architecture that:

1. we experimentally demonstrate for the first time, to the best of our knowledge
2. was fabricated using an industry-standard silicon photonics platform with a single waveguide integrated on-chip photodetector and thermo-optic switches
3. is a fully-packaged photonic chip with standard FC/PC fiber connectors and on-board analog and digital electronics for standalone operation
4. has been shown to retrieve high resolution spectra beyond the Rayleigh criterion and high optical throughput, obtaining the multiplex advantage (Fellgett's advantage)

In addition, through our dFT spectrometer demonstration we describe a unique method for tuning the power of each thermo-optic switch such that light is completely switched into an upper or lower path. This is all shown in our manuscript in addition to our work on spectrum reconstruction using our elastic- D_1 regularized regression method.

Secondly, our work shows a number of obvious differences with reference [1], most notably the overall chip architecture and the intended function. We do acknowledge that there have been several previous demonstrations of variable delay lines and optical switch networks combined to perform different operations. However, we would like to emphasize that the originality of our work encompasses the demonstration of a reconfigurable *interferometer* with the following critical characteristics:

1. all light is collected by a single element photodetector in such a way that the multiplex advantage is obtained
2. each switch state corresponds to a unique interferometer state, such that the number of states (and interferogram measurements) scales exponentially with the number of switches

These key characteristics are not demonstrated or mentioned in reference [1], and as such we believe our work is substantially different from other demonstrations of variable optical delay lines.

Reference [2] noted by the reviewer above refers to a Fourier-transform spectrometer with physical path length differences to also retrieve high-resolution spectra. We have revised our manuscript to include this reference. This theoretical paper depicts an interferometer with several distinct disadvantages with respect to our experimentally validated design:

1. The interferometer in [2] requires N switching stages for N measurements of the interferogram, which will result in a large required chip area to accommodate high-resolution measurements (especially if $N > 64$) and an insertion loss that scales linearly with N . In contrast, our design's chip space and insertion loss scales logarithmically ($\sim \log(N)$).
2. The interferometer in [2] requires N photodetectors and $2N$ switches for N measurements of the interferogram. This makes acquiring high-resolution spectra ($N > 64$) difficult due to the sheer number of analog switches and photodetectors that need to be individually addressed with dedicated control electronics. In contrast, our design requires only modest electronics to address a single photodetector and a smaller number of switches ($\sim \log(N)$). As mentioned in the manuscript, a 1024 channel on-chip dFT spectrometer would require only 10 analog heater controllers and a single photodetector readout circuit.

Furthermore, the reviewer mentions a recent book chapter [3] (published in April 26, 2018) which describes a design similar to that in [2] with the aforementioned disadvantages. The broadband spectrometer depicted in this chapter has a non-unique set of interferometer states for each switch state, such that there can be at most N interferogram measurements (rather than 2^N measurements) for N switches (which incurs a dramatic performance penalty).

[4] Kita, D., Lin, H., Agarwal, A., Richardson, K., Luzinov, I., Gu, T., & Hu, J. (2017). On-chip infrared spectroscopic sensing: redefining the benefits of scaling. *IEEE Journal of Selected Topics in Quantum Electronics*, 23(2). <https://doi.org/10.1109/JSTQE.2016.2609142>

2. What are the dimensions of the optical waveguides used in the spectrometer layout? Rib/ridge waveguide or channel waveguide used? Why did you specifically use silicon material platform? Compactness is an important aspect, however considering the penalties of Silicon photonics in terms of loss and back reflections due to sidewall roughness, it does not give the best performance. The motivation of waveguide/material platform as well as some crucial information that could make this research repeatable for other researchers are missing. Is this due to the fact that there is a commercial interest in this device?

Response:

We used the semi-standard strip silicon optical waveguide design which is 450 nm wide and 220 nm thick. Both rib/ridge and channel waveguides are used for different components. We used the ISIPP25G active silicon photonics MPW service at IMEC, which includes several waveguide types for operation in the telecom C-band including the strip waveguide design we used. We specifically chose silicon as this material platform offers compactness (due to high index contrast), verified propagation loss values (that did not significantly deter from the performance of our spectrometer), efficient & small footprint thermo-optic phase shifters (created by altering the silicon doping profile near the waveguide), and the opportunity for waveguide integrated germanium photodetectors. We found that sidewall induced roughness backscattering and loss did not actually substantially affect the performance of our device. Since all of the components in this particular spectrometer design come from standard Si photonic PDK components as well as prior literature [5], we believe the work can be quite easily reproduced by other researchers (per Reviewer #3's comment).

The demonstrated dFT spectrometer architecture is generic, and can also be readily made with other material platforms (Ge on Si, SiN on SiO₂, etc.) so long as phase modulation mechanisms and appropriately integrated photodetectors are available. We believe this work will motivate future dFT spectrometers in other material platforms and at other wavelength ranges of interest.

[5] Harris, N. C., Ma, Y., Mower, J., Baehr-jones, T., Englund, D., Hochberg, M., & Galland, C. (2014). Efficient, compact and low loss thermo-optic phase shifter in silicon. *Optics Express*, 22(9), 83–85. <https://doi.org/10.1364/OE.22.010487>

3. What are the sources of 2.75 dB loss per stage? Please also talk about the possible improvements if there is any.

Response:

The 2.75 dB of loss that we previously reported was an estimate from the reported insertion loss of the 2×2 multimode interferometers (directly quoted from the ISIPP25G component library), the propagation loss of the single mode waveguides, and an estimate of the loss from custom thermo-optic phase modulators. We further performed a set of elaborate experimental tests to

quantify the insertion loss per switching stage (which is now detailed in the Supplementary Information), and found that the loss is actually 1.69 ± 0.45 dB. Improved component designs as well as improvements to device fabrication will definitely lead to lower overall losses. The Supplemental Information has been appended with an additional section that discusses our insertion loss measurements and future improvements to reduce the total insertion loss across the entire spectrometer following the reviewer's suggestion.

4. How do your thermo-optic switch, 1x2 splitter, 2x1 combiner look like? Please include drawings together with some simulation results to show how much loss is expected in theory from these components. I am assuming that you used Y-splitters mainly, however did you reduce their losses by optimizing the geometry if not a e-beam writing method was used in the fabrication?

Response:

The 1×2 multimode interferometer (used for both splitter/combiner) and the 2×2 multimode interferometer are both standard PDK components for the IMEC ISIPP25G active silicon photonics line. For specifics regarding the 1×2 MMI and 2×2 MMI designs, the reviewer can request this information from europractice-ic (it is technically intellectual property owned by europractice-ic, and a non-disclosure agreement (NDA) is typically required for this information – that is why we are not permitted to list the information in the manuscript; however, europractice-ic MPW users would be granted access to the component library once an NDA is signed). This being said, the designs have all been extensively optimized with respect to loss values, power imbalance, and phase accuracy that were experimentally measured by separate fabrication runs with many test structures.

The thermo-optic phase shifters were custom components made by gradually varying the doping profile near the core of silicon rib waveguides. The design we used, along with accompanying graphics and measurements, is discussed in detail in [5]. In this work, they experimentally measured the insertion loss to be 0.23 ± 0.13 dB, the phase shifting efficiency to be 24.77 ± 0.43 mW/ π . The only difference is the fabrication facility used, and the length of the device (their phase shifter lengths are 61.6 μm and ours are 89.5 μm).

[5] Harris, N. C., Ma, Y., Mower, J., Baehr-jones, T., Englund, D., Hochberg, M., & Galland, C. (2014). Efficient, compact and low loss thermo-optic phase shifter in silicon. *Optics Express*, 22(9), 83–85. <https://doi.org/10.1364/OE.22.010487>

5. Even though the layout given here looks simple, I believe that finding the right OPL values by perturbing the phase values of each thermo-optic switch is laborious. What is the typical time needed to reconstruct spectral information, such as, given in Fig. 4?

Response:

Each thermo-optic switch is only driven at two powers: one which provides a necessary phase-shift to guide light to the “UP” arm, and another which provides the necessary phase-shift to guide light to the “DOWN” arm. We describe in the main text our method of calibrating the powers for each thermo-optic switch (which involves monitoring a top and bottom ‘tap’ port and

perturbing the heater powers until the output spectrum is flat). This calibration process is rather fast (it takes approximately 20 minutes to manually calibrate all 6 switch powers for the 64-channel device – the process can be readily automated with much reduced calibration time), and it need only be performed once. Our control electronics and software is designed to remember the heater powers and so during a spectrum measurement no calibration or manual tuning is necessary.

In our laboratory setup, each interferogram measurement takes 2.7 seconds (this is the total time to switch through all 64 interferometer states). This value is largely limited by the USB latency for controlling and reading values from the DAQ board. To determine the true speed of our spectrometer, we used a function generator and oscilloscope to measure the 10%-to-90% rise time of the thermo-optic switches, which was 30.8 μ s. Thus, a single spectrum could be measured in 1.97 msec.

Reviewer #3 (Remarks to the Author):

This paper demonstrates for the first time the implementation of an on-chip Fourier-transform spectrometer using integrated thermo-optic switches, which are used to reconfigure the circuit to have different optical path lengths at different times. Because of the size, weight, and power advantages of on-chip spectrometers compared to bench top systems they have the potential to find application in many areas.

The critical advantage of this new spectrometer compared to existing on-chip spectrometer designs is that this architecture is easily scalable to very high spectral channel counts with limited corresponding increase of the on-chip footprint of the circuit, which is extremely important if on-chip spectrometers are to be used widely. There are several additional advances presented here, including presentation of a novel spectral retrieval procedure, integration of the spectrometer with on-chip detectors, and packaging with interface electronics.

I believe that this work can have a high impact in the field, that it is technically strong, and that evidence is provided for most of its conclusions. A good level of detail is provided, and it should be possible for another researcher to reproduce this work.

I have several questions and comments that I would like to see addressed before publication:

1. On page 2, paragraph 2 it is stated that:

“Furthermore, prior work has demonstrated Fourier-transform spectrometers using arrays of discrete Mach-Zehnder interferometers (MZIs)^{15,18,25,26}, although these approaches are not practical for large spectral channel counts (due to excessive chip footprints) and **suffer from the same SNR penalties as dispersive-type spectrometers since power is evenly split into each of the N spectral channels.**”

The bold part of the sentence is potentially misleading, as it has previously been argued that for these types of devices (e.g. Florjanczyk et al. “Multiaperture planar waveguide spectrometer formed by arrayed Mach-Zehnder interferometers” Optics Express, vol. 15, issue 26, 2007) a single input waveguide is used only to simplify device characterisation, and that a separate input waveguide could be used for each MZI. This would avoid the SNR penalties from splitting into

N spectral channels, and would in fact increase the etendue, allowing more light to be coupled into the chip from spatially extended sources. Please could the bold text be amended?

Response:

We thank the reviewer for their comments and feedback. We have removed the statement following the reviewer's suggestion.

2. What is the optical insertion loss of the spectrometer, and how much does it vary across the different interferometer channels?

Response:

We performed a set of experiments (explained in detail in the updated Supplementary Information) to determine the insertion loss of the 64-channel spectrometer, as well as the insertion loss per switching stage. We found that the 64-channel spectrometer has a 9.08 ± 1.72 dB insertion loss, averaged across all wavelengths in the 1550 nm – 1570 nm band considered in this manuscript. The insertion loss per stage, determined by a linear regression of the losses for a 1-stage dFT, 2-stage dFT, and 3-stage dFT spectrometer was determined to be 1.69 ± 0.45 dB. As for the variation of the total insertion loss amongst different channels (or “switch-states” of the spectrometer), this value is more challenging to quantify due to the different wavelength-dependent interference fringes for each switch state. However, we did measure the average loss and maximum loss across the entire 20 nm band for each channel, and we found that amongst the 64 channels the spectrum-averaged loss has a standard deviation of 0.208 dB, and the maximum loss (across the spectrum) has a standard deviation of 0.283 dB.

3. Could the authors please comment on the visibility (i.e. extinction ratio) of the interferometer fringes, and on what the consequences are for the sensitivity or dynamic range of the spectrometer?

From our measurements and analysis, we found that the average extinction ratio across all fringes and all interferometer channels is 21 ± 3 dB, and we have added this to the main text. Ultimately, the visibility is limited by the difference in losses between the two interferometer arms. We don't anticipate the sensitivity or dynamic range of the spectrometer will be significantly impacted by suboptimal visibility, but rather by the sensitivity of the photodetector used and insertion loss across the entire device. The quality of the spectrum reconstruction would be expected to decrease for an interferometer with poor visibility, which is the consequence of having a less orthogonal set of basis vectors by which to reconstruct the spectrum.

4. How long does the measurement of a single spectrum take (excluding additional averaging time)?

Response:

In our laboratory setup, each interferogram measurement takes 2.7 seconds (this is the total time to switch through all 64 interferometer states). This value is largely limited by the USB latency for controlling and reading values from our data acquisition (DAQ) board. To determine the true speed of our spectrometer, we used a function generator and oscilloscope to measure the 10%-to-

90% rise time of the thermo-optic switches, which was 30.8 μs . Thus, a single spectrum could be measured in 1.97 msec.

5. Since for some applications a wide spectrometer bandwidth would be crucially important, could comments be added to the paper to explain what limits the bandwidth of this current device, as well as what are the fundamental bandwidth limitations of a device of this kind?

Response:

For a free-space Fourier-transform spectrometer (Mach-Zehnder), the bandwidth is traditionally limited by the transparency window of the beam-splitter and the responsivity curve of the photodetector. In the dFT spectrometer case, the fundamental limitation of the bandwidth is likewise the transparency window of the core and cladding (silicon and silicon dioxide in our case) and the responsivity of the photodetector (Ge in our case), as well as single-mode condition of the input waveguide (which sets a lower bound for its operation wavelength). Additionally, our spectrometer uses fiber grating couplers, which are inherently narrow-band (limited to $\sim 100\text{nm}$), but this can be alleviated with edge coupling schemes.

Other integrated components, such as the 1×2 MMI, 2×2 MMI, and thermo-optic phase shifters all have some broad wavelength-dependent response. This is typically a problem that requires careful corrections when using Fourier transforms to compute the input light spectrum. However, since we measure the full calibration matrix and use this to determine the spectrum, we care only about the degree of orthogonality of our basis vectors. We intend to show, in follow-up publications, that our dFT architecture can be applied to broadband applications covering over half an octave.

6. On p. 4, paragraph 1, it is stated that “An additional benefit is that the device is far less sensitive to temperature variations than existing on-chip FTIR spectrometers, as the temperature-induced OPL fluctuation scales linearly with the physical length of the interferometer arms”.

Could the authors please be more specific about which types of existing spectrometers they are referring to? It is my understanding that for a specific resolution and bandwidth the device presented here would require the same OPL as MZI array type devices (e.g. ref 2, 15, 18), and that therefore the sensitivity to thermal variations would be the same. Would any strategies be required in future to compensate for thermal variations?

Response:

We thank the reviewer for this comment. This question is similar to the question asked by Reviewer 1, so we would like to refer Reviewer 3 to this answer for further details.

We believe that temperature compensation is readily possible for a device of this nature, so long as the appropriate basis is measured at multiple experimentally relevant temperatures. One method we have discussed is measuring the basis at a sufficient number of temperatures, interpolating the temperature dependence of each matrix element, and using this to perform the reconstruction. In this scenario, the temperature need only be read once from an on-chip or near-chip thermistor (prior to measuring the interferogram), and the corresponding matrix would be chosen to perform the reconstruction. The advantage of using this technique is that the system-

level complexity is reduced (no temperature controlled stage and PID control), but this comes at the cost of a long initial calibration step and increased on-board memory requirements (for storing the interpolated matrix).

7. Please explain more clearly the difference between the “Average fixed parameter compute time” and “Hyper-parameter search time” in the caption to Table 2.

Response:

“Average fixed parameter compute time” is the average time it takes for the computer to solve for the spectrum given an interferogram and a set of hyperparameters. The “hyperparameter search time” is the time spent searching for optimal hyperparameters (values that minimize $R^2(A_2x, y_2)$) and is equal to the average fixed parameter compute time multiplied by the number of hyperparameters searched. So if it takes 1 msec for the fixed parameter compute time, and the search space is 200×200 , the total “hyperparameter search time” is $200 \cdot 200 \cdot 1\text{msec} = 40\text{sec}$. Once the optimal hyperparameters are determined, the actual spectrum reconstruction takes a time equal to the “fixed parameter compute time” to calculate the spectrum.

We would like to point out a few subtleties:

1. The reported “hyperparameter search time” assumes we perform all calculations in *serial*, but in practice this is an “embarrassingly parallel” computational problem, meaning it can be trivially computed on multiple processor cores or GPUs at once (since the calculations don’t need to talk with one another). This would reduce the search time by the number of processors.
2. We also perform a brute-force optimization across the entire hyperparameter search space, but in practice a gradient-descent search method (or other appropriate nonlinear optimization method) would considerably reduce the hyperparameter search time.

For the reasons above, we have decided that perhaps the “hyperparameter search time” is not as relevant of a metric for our dFT spectrometer system or algorithm (rather, it is dependent on the implementation). We have eliminated this column from Table 2, as shown in the revised manuscript. The values previously in this column can be obtained by multiplying the fixed parameter compute time by the hyperparameter search space (for our particular implementation).

8. Since compute times are given for the different reconstruction methods, please provide some details of the computer that was used for these calculations.

Response:

For our spectrum reconstruction, we used a laptop with an Intel Xeon E3-1505M v5 CPU and 16 GB of RAM (though none of the computations required significant memory usage). This text has been added to the caption of Table 2 in the main text for clarity.

9. How much variation is there in the R^2 values given in table 2 when the reconstruction methods are tested on different broadband spectra (e.g. from figures 4A and 4B) for the broad spectra, or when the lasers are tuned to different wavelengths for the sparse spectra? In other words, are the R^2 values presented in table 2 the “best case” for the Elastic-D1 method? Would average

R² values for different input spectra be more appropriate?

Response:

Thank you for this detailed comment. It is true that the R² value is a strong function of the type of input spectrum (especially for traditional compressive sensing techniques like LASSO and BPDN). To provide the reader with a more comprehensive picture of the quality of reconstructions offered by each technique, we have updated Table 2 with R² values for all three broadband spectra (Figures 4A, 4B, and 4C). In addition, we have performed additional measurements of a single-frequency laser at 11 different wavelengths, and provided the average and standard deviation of the R² values for these similarly sparse spectra.

10. In the supplementary materials, on page 1, paragraph 5, it is stated that "...we use a standard holdout cross-validation technique, which requires only two successive measurements of the interferogram, characterized by the same input signal with different noise."

Would taking more than two successive measurements improve the accuracy of the calibration?

Response:

In general, the reviewer is correct to note that more than two successive measurements of the interferogram should improve the accuracy of the ultimate reconstruction (spectrum vector). The subtlety is that there are then many different ways of using this extra data to improve the accuracy of the solution. A straightforward method for doing this for N interferogram measurements (denoted by y_i) would be to find the hyperparameters that maximize the sum of *all* coefficients of determination:

$$\sum_i^{N-1} R^2(A_2 x_i, y_N)$$

where x_i is the computed spectrum from A_1 and y_i . In this way, hyperparameters will be chosen that are the most robust to experimental noise.

We should note that there are also several other methods of using this extra information that could make the reconstruction in principle worse. For example, using multiple interferogram measurements to determine multiple different values for each hyperparameter, and then taking the average would not guarantee a more accurate hyperparameter value (for the same reason that averaging two different local maxima positions does not guarantee a position with a higher value).

REVIEWERS' COMMENTS:

Reviewer #1 (Remarks to the Author):

The authors answers the different point raised by the reviewers. I believe the paper can be published.

Reviewer #2 (Remarks to the Author):

Thanks to the authors for their detailed and satisfactory explanations for the comments/remarks I made. My decision on this paper is "Accepted". Congratulations and all the best!

Reviewer #3 (Remarks to the Author):

I am satisfied that the authors have fully and in detail addressed all of the questions and comments raised in the previous round of reviews, and have modified the manuscript accordingly.

Response to Reviewer Comments NCOMMS-18-13737

We would like to thank all of the reviewers for their insightful questions and comments during this review process.

Reviewer #1 (Remarks to the Author):

The authors answers the different point raised by the reviewers. I believe the paper can be published.

Reviewer #2 (Remarks to the Author):

Thanks to the authors for their detailed and satisfactory explanations for the comments/remarks I made. My decision on this paper is "Accepted". Congratulations and all the best!

Reviewer #3 (Remarks to the Author):

I am satisfied that the authors have fully and in detail addressed all of the questions and comments raised in the previous round of reviews, and have modified the manuscript accordingly.